# Turning Drift into Constraint: Robust Reasoning Alignment in Non-Stationary Multi-Stream Environments

Xiaoyu Yang [1]   En Yu [1]   Wei Duan [1]   Jie Lu [1]

## Abstract

This paper identifies a critical yet underexplored challenge in reasoning alignment from multiple multi-modal large language models (MLLMs): In non-stationary environments, the diverse reasoning distributions of source models often evolve unpredictably, transmitting systematic biases and drift to the target model. To address this, we formulate multi-source reasoning alignment as a constraint satisfaction problem under concept drift theory. We propose Autonomous Preference Optimization (APO), a novel framework that treats inter-model divergences not as noise, but as dynamic negative constraints. APO operates via a two-stage protocol: first, supervised bootstrapping projects the target model into the capability union of source models; second, constraint-aware optimization synthesizes a consistent consensus manifold by explicitly suppressing drifting trajectories via a multi-negative Plackett-Luce objective. Extensive experiments on chest X-ray interpretation demonstrate that our 7B model achieves superior robustness, outperforming even proprietary source models in average accuracy. Furthermore, we release CXR-MAX, a large-scale benchmark comprising 170,982 reasoning trajectories from seven large-scale MLLMs to facilitate research on reasoning alignment under drift. Code and data are available at: https://github.com/XiaoyuYoung/APO.

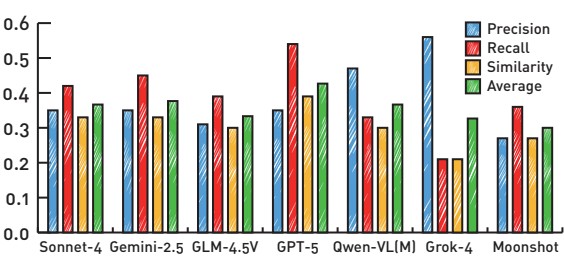

*(a)* Concept drift among source MLLMs.

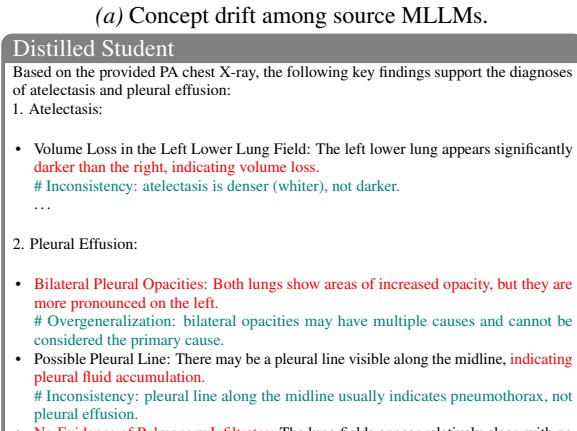

*(b)* Example of drift-biased target model. Red markers pinpoint specific flaws in the generated reports, with green markers providing the rationale behind these errors.

*Figure 1.* Transmission of concept drift behind alignment of MLLMs.

## 1. Introduction

Recent advancements in Large Language Models (LLMs) have shifted the paradigm from training isolated models to aligning with the collective intelligence of multiple existing models (Dai et al., 2025; Wan et al., 2024; Saha et al., 2023). Leveraging diverse reasoning priors from multiple source models has proven effective in complex tasks such as visual question answering in specialized domains (e.g., medical diagnosis) (Shen et al., 2025; Gui et al., 2024), while also enhancing the generalization of chain-of-thought (CoT) reasoning (Feng et al., 2025b; Shu et al., 2025; Cao et al., 2025a). Furthermore, reasoning fusion strategies and personalized explanation alignment demonstrate that integrating complementary expertise significantly boosts target model performance. As noted in recent surveys (Fang

[1]Australian Artificial Intelligence Institute (AAII), Faulty of Engineering and Information Technology, University of Technology Sydney, Australia. . Correspondence to: Jie Lu <jie.lu@uts.edu.au>.

*Proceedings of the 43rd International Conference on Machine Learning*, Seoul, South Korea. PMLR 306, 2026. Copyright 2026 by the author(s).

et al., 2025), leveraging multiple large models as reference streams has emerged as a standard paradigm for efficient capability acquisition.

However, aligning with multiple models introduces a critical yet often overlooked challenge: the sources are fundamentally non-stationary. Unlike static environments, the reasoning trajectories generated by different source models exhibit significant inter-model drift, i.e., divergent distribution shifts arising from varying pre-training biases and architectural differences. Concept drift theory (Lu et al., 2019; Yang et al., 2025a) offers a compelling analytical lens to examine these dynamics. From this perspective, the target model is exposed to a multi-stream environment where reasoning paths may asynchronously converge, diverge, or directly conflict. Naive alignment strategies that indiscriminately absorb these heterogeneous streams risk inducing concept misalignment, causing the target model to internalize contradictory logic and ultimately leading to catastrophic error propagation and reduced robustness in safety-critical scenarios.

To systematically characterize these dynamics, we analyzed the reasoning trajectories generated by diverse source MLLMs on the MIMIC-CXR benchmark within the concept drift framework, as shown in Figure 1. Our empirical investigation reveals fundamental characteristics in multi-stream drift. First, distinct source models exhibit complementary divergence:

*Observation* 1.1. *While some models, such as Qwen-VL-Max, adhere to high-precision, concise reasoning distributions, others like GPT-4o favor high-recall, expansive elaboration. This suggests that the "true" reasoning manifold lies within the consensus of these divergent streams, rather than in any single trajectory.*

Second, naive alignment leads to distributional corruption:

*Observation* 1.2. *The target model trained simply to mimic these drifting streams does not automatically synthesize their strengths; instead, it internalizes the union of their biases, resulting in hallucinations and semantic inconsistencies.*

Crucially, these observations lead to a pivotal insight: the drifting regions, where source models significantly disagree, should not be merely treated as noise to be averaged out. Instead, they serve as explicit negative constraints that delineate the decision boundaries of robust reasoning. This perspective transforms the alignment problem from simple imitation to a constraint-satisfaction process, where the model learns what to avoid (drift) as effectively as what to follow (consensus).

Therefore, synthesizing the above findings, we are confronted with a fundamental dilemma in multi-stream integration: the very diversity that enhances collective reasoning also introduces non-stationary drifts. This necessitates a paradigm shift from passive aggregation to active constraint satisfaction, raising the core research question of this work:

***How can we autonomously turn drift into constraint, thereby achieving robust reasoning alignment in non-stationary environments?***

Guided by this constraint-centric perspective, we propose Autonomous Preference Optimization (APO), a framework designed to operationalize the *drift-as-constraint* insight through a rigorous three-stage alignment protocol. Initially, the target model is exposed to diverse reasoning streams to acquire a broad coverage of domain capabilities, establishing a foundational but noisy capability space. Second, instead of passive imitation, the model then aggregates these streams to synthesize a consensus manifold, a self-consistent trajectory that resolves inter-model conflicts and mitigates individual hallucinations. In the final phase, we reformulate the alignment objective by treating the synthesized consensus as the positive reference and the divergent, drifting trajectories as negative constraints. By maximizing the likelihood of the consensus manifold while actively suppressing the probability of drifting patterns, APO effectively utilizes their own conflicts among source models to sharpen the decision boundaries, achieving robust alignment without reliance on ground-truth supervision.

In summary, our work advances the field of robust model alignment through the following contributions:

- We establish a novel framework that recasts multi-source reasoning integration as a constraint satisfaction problem in non-stationary environments. Within the perspective of concept drift theory, we demonstrate how conflicting reasoning trajectories can be transformed from disruptive noise into actionable negative constraints for decision boundary sharpening.

- We propose Autonomous Preference Optimization (APO), a self-supervised alignment strategy that eliminates the need for ground-truth labels. By treating the consensus among source models as positive signals and their drifting conflicts as negative constraints, APO autonomously constructs preference pairs to guide robust reasoning alignment.

- We conduct extensive evaluations across diverse benchmarks. Our results demonstrate that APO achieves superior robustness and generalization while utilizing only 10% of the data typically required by standard alignment methods, effectively mitigating drifts inherent in individual source models.

- To facilitate future research on alignment under drift, we release CXR-MAX, a large-scale benchmark comprising over 170k reasoning trajectories with fine-

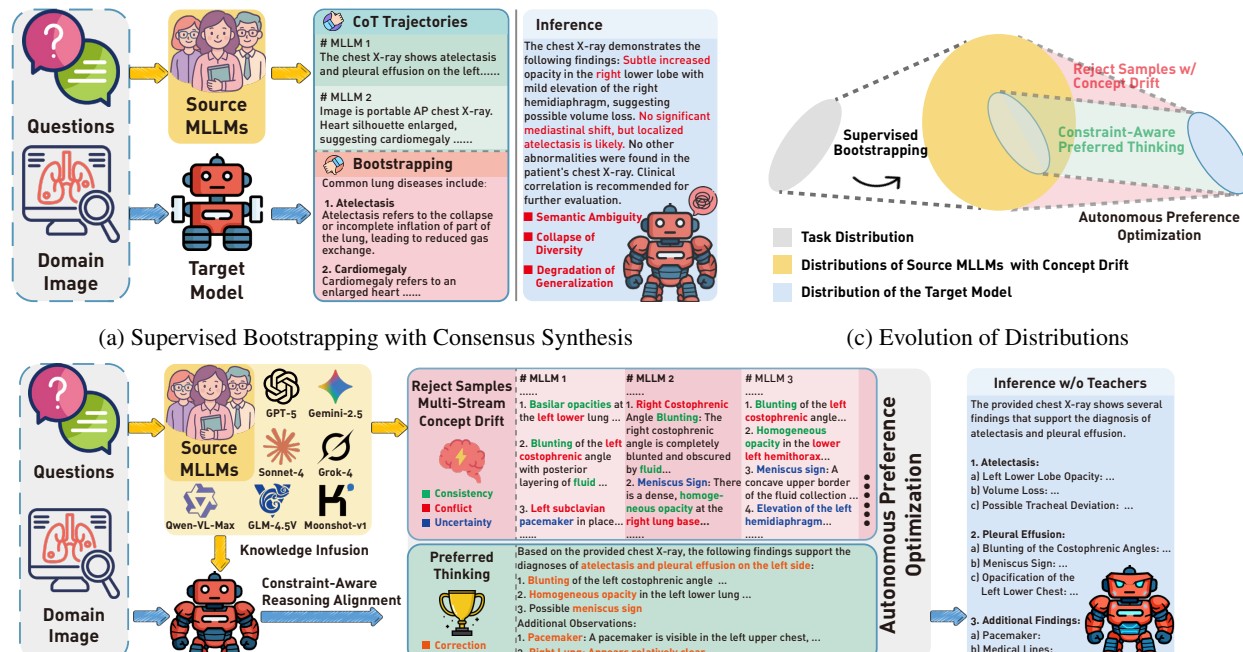

*Figure 2.* The main contributions of our methods. (a) **Supervised Bootstrapping with Consensus Synthesis.** In the initial phase, the target model undergoes Supervised Bootstrapping to establish a broad capability covering by assimilating the collective knowledge of source MLLMs. However, as shown in the inference block, this naive integration inevitably inherits non-stationary inter-model drift, resulting in hallucinations and semantic ambiguities. (b) **Constraint-Aware Optimization for Robust Reasoning Alignment.** To mitigate the inherited drift, we propose a Constraint-Aware protocol. The model first employs in-context extraction to synthesize self-consistent consensus trajectories as preferred thinking. Crucially, rather than simply discarding the conflicting source outputs, APO repurposes them as negative constraints within a Plackett-Luce preference formulation, explicitly suppressing the probability of generating drifting patterns. (c) **Evolution of Distributions.** The distributional dynamics of our alignment process. Initially, bootstrapping projects the target model into the union of source distributions (Yellow). Subsequently, APO refines this space by treating the drifting regions (Red) as decision boundaries to be avoided, effectively carving out a robust consensus manifold (Green) for reliable reasoning.

grained alignment annotations. This serves as a critical testbed for studying inter-model dynamics and reasoning consistency in high-stakes domains.

## 2. Methodology

In this section, we first present the theoretical formulation of multi-stream reasoning dynamics. Subsequently, we introduce **Autonomous Preference Optimization (APO)**. Our framework recasts the alignment challenge as a constraint satisfaction problem, following a two-stage protocol: Supervised Bootstrapping with Consensus Synthesis, and Constraint-Aware Optimization.

### 2.1. Modeling Non-Stationary Reasoning Drift in Multi-Stream Alignment

In this section, we extend the theoretical framework of concept drift to the setting of multi-source MLLMs alignment. We posit that the divergence among source models is not a static error margin but a dynamic, non-stationary process. Specifically, we map the autoregressive reasoning steps of

the chain-of-thought to the temporal dimension in traditional drift theory, emphasizing the unpredictable distributional shifts that arise as the reasoning trajectory unfolds.

Prior studies on concept drift predominantly address single-stream inference (Yang et al., 2025b; 2026b), where an individual source model $\pi$ autoregressively generates the token at position $j$, conditioned on the visual input $v$ and textual prompt $l$. Thus, the partial token sequence $t_{<j}$ of the CoT trajectory is given by

$$t_j \sim \pi(\cdot \mid v, l, t_{<j}). \tag{1}$$

Thus, the single-stream process is formalized as follows:

***Definition 2.1.*** *(**Single-Stream Reasoning State**) The autoregressive reasoning trajectory of a single source MLLM unfolds as a sequential stream $S = \{s_0, \ldots, s_L\}$, where each state $s_j = (t_{<j}, z_j)$ comprises the partial token sequence $t_{<j}$ generated up to step $j$ and the corresponding latent predictive distribution $z_j = \pi(\cdot|v, l, t_{<j})$ that governs the subsequent generation.*

Building on this formulation, we extend the framework to

a multi-source setting, where the target model operates in an environment composed of $N$ distinct reasoning streams. Unlike static ensembles where member disagreement is constant, the correlation and conflict among source MLLMs evolve dynamically as the reasoning deepens. Formally, we define this as multi-stream reasoning drift:

***Definition* 2.2. (*Multi-Stream Reasoning Drift*)** *Consider $N$ CoT streams corresponding to $N$ source models. Let the collective state at reasoning step $j$ be denoted by $\mathcal{S}_j = (s_j^1, \ldots, s_j^N)$, where $s_j^u$ represents the state of the $u$-th source model. We define the reasoning alignment process as experiencing concept drift if the joint distribution of the collective states evolves non-stationarily across steps. That is, for any two distinct reasoning steps $j$ and $j + \Delta$, the joint probability distributions differ:*

$$P_j(\mathcal{S}) \neq P_{j+\Delta}(\mathcal{S}). \tag{2}$$

Assuming that source models generate reasoning trajectories independently conditioned on the input, that they are trained independently without mutual fine-tuning, the joint distribution $P_j(\mathcal{S}_j)$ at step $j$ can be factorized into the product of marginal distributions:

$$P_j(\mathcal{S}_j) = \prod_{u=1}^{N} P(t_{<j}^u | v, l) \cdot P(z_j^u | t_{<j}^u, v, l). \tag{3}$$

Eq. (3) highlights the characteristics of the drift in reasoning alignment. The term $\prod P(t_{<j}^u)$ represents the accumulated historical divergence, while $\prod P(z_j^u | \cdot)$ represents the instantaneous reasoning drift. By framing this as concept drift, we capture the unpredictable nature of the alignment landscape: at step $j$, source models might converge on an inference result, but at step $j + \Delta$, they may diverge wildly in their rationale. This dynamic variation creates a non-stationary supervision signal for the target model, necessitating an alignment strategy that adapts to these evolving distributional discrepancies rather than treating them as static noise.

## 2.2. Supervised Bootstrapping with Consensus Synthesis

Building on the formulation of non-stationary reasoning dynamics in Eq. (3), we identify a critical challenge: the intrinsic inconsistencies and biases in source models, if naively aligned, propagate to the target model as systematic errors, as demonstrated in Observation 1.2. To address this, we propose a two-stage protocol: first, bootstrapping the target model to cover the collective capabilities of the sources, and second, extracting a consistent reasoning trajectory to resolve inter-model drift.

The target model $\pi_\theta$ first undergoes a supervised bootstrapping phase. Despite the presence of drift, the goal here is to project the target model into the union of the source

models' representational spaces, ensuring a comprehensive capability covering. Specifically, at each reasoning step, the source models provide a mixture of predictive distributions. We formulate the objective as minimizing the collective divergence between the initial model $\pi_{\text{init}}$ and the ensemble of distributions over source MLLMs $\{\pi_u\}_{u=1}^N$. The optimal aligned distribution $q^*$ is defined as:

$$\begin{aligned} &q^*(z|t_{<j}) \\ &= \arg\min_q \sum_{u=1}^{N} \mathbb{D}_{\text{KL}} \left( \pi_u(\cdot|t_{<j}, v, l) \,\|\, q(\cdot|t_{<j}) \right), \end{aligned} \tag{4}$$

where $q^*$ denotes the optimal aligned distribution that encapsulates the collective knowledge of all source MLLMs within the target model. Upon convergence, we denote the resulting bootstrapped model as $\hat{\pi}_{\text{st}}$. Through this bootstrapping process, the bootstrapped model $\hat{\pi}_{\text{st}}$ assimilates the heterogeneous knowledge, reconciling conflicting signals not by adhering to a single source, but by establishing a foundational feature space that encapsulates the collective expertise of the source ensemble.

While the bootstrapped model $\hat{\pi}_{\text{st}}$ has acquired broad domain capabilities, it remains susceptible to drift. The subsequent step addresses this by leveraging the model's own emergent reasoning capabilities to extract the consensus manifold from the noisy source outputs. We employ an in-context extraction strategy. The original reasoning trajectories $\mathcal{T} = \{\tau^1, \ldots, \tau^N\}$ generated by various source models are aggregated for the same instance. These trajectories serve as a noisy context containing both valid signals and drifting errors. We then condition the target model on this context to generate a refined self-consistent trajectory $t^+$:

$$t^+ \sim \hat{\pi}_{\text{st}}(\cdot \mid v, l, \text{Context} = \mathcal{T}). \tag{5}$$

By conditioning on the concatenated observations of inter-model drift $\mathcal{T}$, the target model acts as a reasoned aggregator. It filters out incoherent drift, i.e., tokens lacking cross-model support, and amplifies the logical intersections, thereby extracting a consensus trajectory $t^+$ that represents the preferred reasoning path. This $t^+$ serves as the anchor for the subsequent optimization phase.

## 2.3. Constraint-Aware Optimization via APO

Having extracted the consensus trajectory $t^+$ in Eq. (5), the final challenge is to enforce this consensus while explicitly suppressing the drifting modes inherent in the source models. The target model must not only learn what to generate (the consensus) but also what to avoid (the inter-model drift). Consequently, we transition from the bootstrapping to the constraint-aware optimization. Here, the extracted consensus $t^+$ serves as the positive signal, while the raw,

conflicting trajectories $\mathcal{T}$ from source models serve as negative constraints. By maximizing the margin between the consensus and the drift, the target model sharpens its decision boundaries against hallucination and variance.

Formally, we frame this as an autonomous preference optimization problem. We employ the bootstrapped model $\hat{\pi}_{st}$ as the reference policy to constrain the deviation of the optimizing policy $\pi_\theta$. The implicit reward function $r(v, l, t)$, derived from the optimal policy assumption in DPO (Rafailov et al., 2023), is defined as:

$$r(v, l, t) = \beta \log \frac{\pi_\theta(t|v,l)}{\hat{\pi}_{st}(t|v,l)}, \qquad (6)$$

where $\beta$ is a parameter controlling the deviation from the base reference policy $\hat{\pi}_{st}$. Under this formulation, we treat the consensus $t^+$ as the preferred solution and the set of drifting source trajectories $\mathcal{T} = \{\tau^1, \ldots, \tau^N\}$ as the dispreferred set. To handle multiple negative constraints simultaneously, we generalize the Bradley-Terry model (Hunter, 2004) to a Plackett-Luce style (Plackett, 1975) preference probability, where the consensus is compared against the ensemble of drifting outputs:

$$
\begin{aligned}
&P(t^+ \succ \mathcal{T}|v,l) \\
&= \frac{\exp(r(v,l,t^+))}{\exp(r(v,l,t^+)) + \sum_{u=1}^{N} \exp(r(v,l,\tau^u))}.
\end{aligned} \qquad (7)
$$

Here, the denominator aggregates the exponential rewards of all drifting trajectories, treating them as competing hypotheses that must be suppressed. The Autonomous Preference Optimization (APO) objective is then to maximize the log-likelihood of this preference probability:

$$\mathcal{L}_{APO} = -\mathbb{E}_{(v,l,t^+,\mathcal{T})} \left[ \log P(t^+ \succ \mathcal{T}|v,l) \right]. \qquad (8)$$

Substituting Eq. (6) and Eq. (7) into Eq. (8), we derive the final gradient-descent objective:

$$
\begin{aligned}
&\mathcal{L}_{APO}(\pi_\theta) = \\
&-\mathbb{E}_{(v,l,t^+,\mathcal{T})} \left[ \log \frac{(\frac{\pi_\theta(t^+|v,l)}{\hat{\pi}_{st}(t^+|v,l)})^\beta}{(\frac{\pi_\theta(t^+|v,l)}{\hat{\pi}_{st}(t^+|v,l)})^\beta + \sum_{u=1}^{N}(\frac{\pi_\theta(\tau^u|v,l)}{\hat{\pi}_{st}(\tau^u|v,l)})^\beta} \right].
\end{aligned}
$$
$$(9)$$

Minimizing $\mathcal{L}_{APO}$ forces the target model $\pi_\theta$ to satisfy two dynamic conditions: (1) increasing the likelihood of the consensus $t^+$ relative to the reference $\hat{\pi}_{st}$, and (2) decreasing the likelihood of the specific drifting patterns $\tau^u$ generated by source models. This effectively transforms the inter-model drift from a source of noise into a source of supervision. By explicitly suppressing the probability mass in the drifting regions of the reasoning space, APO carves out a robust manifold for reliable reasoning, achieving alignment without external ground-truth supervision.

**Remark** 2.3. *(Distinction from DPO) While APO leverages the theoretical objective of DPO, it fundamentally diverges in three key aspects:*

**Endogenous Preference Construction:** *Unlike standard DPO, which relies on static, external annotations, APO is autonomous. It dynamically constructs supervision signals by treating the synthesized consensus as the positive reference and the specific drifting modes of source models as negative constraints.*

**Multi-Constraint Topology:** *APO generalizes the pairwise ranking loss to a multi-negative Plackett-Luce formulation. This transforms the alignment problem into a constraint satisfaction task, enforcing the suppression of multiple divergent trajectories simultaneously.*

**Active Unlearning Objective:** *Rather than merely maximizing a generic reward, APO explicitly targets the active unlearning of heterogeneous biases inherent in non-stationary environments, a capability critical for robust multi-stream alignment.*

### 2.4. CXR-MAX Dataset for Reasoning Alignment

To evaluate reasoning alignment in non-stationary environments, a dataset exhibiting high-variance inter-model drift is essential. However, existing benchmarks typically rely on single-source annotations or static consensus, failing to capture the dynamic conflicts inherent in multi-stream reasoning. Addressing this gap, we introduce **CXR-MAX** (**M**ulti-source **A**lignment for **X**-rays), a large-scale benchmark designed to facilitate the study of autonomous preference optimization in high-stakes domains.

CXR-MAX extends the MIMIC-CXR dataset (Johnson et al., 2019) by aggregating reasoning trajectories from seven distinct, publicly available MLLMs. CXR-MAX provides 170,982 distillation instances of reasoning trajectories covering 14 thoracic pathologies, establishing a large-scale benchmark for reasoning alignment with multiple reasoning trajectories from various MLLMs in clinical chest X-ray interpretation. Additional details are provided in Appendix B.

## 3. Experiments

In this section, we verify the robustness, consistency and generalization of our proposed autonomous distillation under non-stationary multi-stream environments.

The MIMIC-CXR dataset (Johnson et al., 2019) serves as an ideal training environment for our method, since medical diagnosis embodies the sophisticated reasoning and

*Table 1.* **Evaluation results of multi-label chest diseases classification on MS-CXR-T.** Top-1 accuracy is applied to evaluate the performance of different methods. The best-performing models are highlighted in red, with the second-best in blue. Comparison methods include CTrans (Bannur et al., 2023b), CheXRel-Net (Karwande et al., 2022), BioViL (Boecking et al., 2022) , BioViL-T (Bannur et al., 2023b) , Med-ST (Yang et al., 2024a), TempA-VLP (Yang & Shen, 2025) and CoCa-CXR (Chen et al., 2025).

| | Venue | Con. | PE | Pna. | Pnx. | Ede. | Avg. |
|---|---|---|---|---|---|---|---|
| *Full Data Training* | | | | | | | |
| CheXRelNet | MICCAI'22 | 0.47 | 0.47 | 0.47 | 0.36 | 0.49 | 0.45 |
| BioViL | ECCV'22 | 0.56 | 0.63 | 0.60 | 0.43 | 0.68 | 0.58 |
| CTrans | CVPR'23 | 0.44 | 0.61 | 0.45 | 0.32 | 0.66 | 0.49 |
| BioViL-T | CVPR'23 | 0.61 | 0.67 | 0.62 | 0.43 | 0.69 | 0.60 |
| Med-ST | ICML'24 | 0.61 | 0.67 | 0.59 | 0.65 | 0.54 | 0.61 |
| TempA-VLP | WACV'25 | 0.65 | 0.59 | 0.73 | 0.43 | 0.77 | 0.64 |
| CoCa-CXR | MICCAI'25 | 0.70 | 0.70 | 0.61 | 0.73 | 0.72 | 0.69 |
| *10% Data Training w/o Radiologist Reports* | | | | | | | |
| **Ours** | **This paper** | **0.84** | **0.67** | **0.78** | **0.96** | **0.65** | **0.78** |

high-stakes practicality that our distillation approach aims to capture. It presents 371,920 chest X-rays associated with 227,943 imaging studies from 65,079 patients. And images are provided with 14 labels with corresponding free-text radiology reports, namely Atelectasis (Ate.), Cardiomegaly (Car.), Consolidation (Con.), Edema (Ede.), Enlarged Cardiomediastinum (ECM), Fracture (Fra.), Lung Lesion (LL), Lung Opacity (LO), Pleural Effusion (PE), Pneumonia (Pna.), Pneumothorax (Pnx.), Pleural Other (PO), Support Devices (SD) and No Finding (NF).

Acknowledging the additional computational overhead and costs associated with employing multiple teachers, we intentionally and deliberately restricted our method to only 1/10 of the whole MIMIC-CXR, underscoring the efficacy of our method in achieving high-quality knowledge transfer from the drifting teachers, even under limited data conditions. The list of chosen random samples is given in our code.

Additionally, we relied solely on the classification labels from MIMIC-CXR and did not utilize the original radiology reports for training. It is motivated by our focus on reasoning alignment from dynamic multiple MLLMs instead of static human annotations, as well as the limited generalizability of human-annotated reports with reasoning trajectories, which are often scarce in the domain-specific area.

In terms of the model, we employ Qwen2.5-VL (7B) (Bai et al., 2025) as the target model to perform supervised bootstrapping and autonomous preference optimization, cascadedly. And they only train one epoch for each stage with a batch size of 2. More detailed experimental implementations are given in Appendix C.

## 3.1. Robust Reasoning Alignment

To rigorously evaluate the robustness of our proposed framework in non-stationary environments, we compare it against state-of-the-art methods on the MS-CXR-T benchmark (Bannur et al., 2023a). A critical distinction in our experimental setup is limited data: while baseline methods utilize the full training set with radiologist reports, our model is trained on only 10% of the data, relying solely on reasoning alignment from drifting source models without ground-truth report supervision.

As presented in Table 1, our approach achieves a remarkable average performance of **0.78**, establishing a new state-of-the-art. Notably, we outperform the second-best method, CoCa-CXR (Chen et al., 2025), by a significant margin of nearly 9%, despite the extreme data scarcity. This result empirically validates our core hypothesis: transforming inter-model drift into negative constraints allows the student to learn more robust decision boundaries than simply imitating ground-truth data.

We achieve dominant scores of 0.96 on pneumothorax (Pnx.) and 0.84 on consolidation (Con.), surpassing the runner-up by 0.23 and 0.14, respectively. It can be attributed to the constraint-aware optimization in APO. Pneumothorax, characterized by subtle pleural lines, often triggers uncertainty in individual source models. By suppressing these drifting uncertainties and reinforcing the consensus, our model sharpens its sensitivity to these critical visual cues.

Besides, while our method trails the top-performing CoCa-CXR by a narrow margin of 0.03 on pleural effusion (PE), we attribute this performance gap to CoCa-CXR's use of additional data from Chest ImaGenome (Wu et al., 2021) in addition to standard MIMIC-CXR. In terms of the edema (Ede.), we argue it is due to the conservative consensus nature of APO. Edema typically presents as diffuse, hazy opacities, causing high drift among source models. Since APO treats high-variance drift as negative constraints to prevent hallucination, the model may adopt a more conservative threshold for such ambiguous classes, trading off some recall for reasoning safety.

Beyond comparing with standard domain-specific baselines, a more rigorous evaluation benchmarks our 7B-parameter target model against the proprietary source MLLMs, which possess vastly superior parameter scales, such as GPT-5 and Claude Sonnet-4. As visualized in Table 2, despite the immense disparity in model size, our approach achieves the highest average accuracy of 0.78 across all diseases, surpassing every single source MLLM. This counter-intuitive result empirically demonstrates that our constraint-aware optimization empowers the compact target model to synthesize a consensus manifold that effectively integrates the diverse strengths of the source ensemble, allowing it to stand on the

*Table 3.* **Evaluation results of diagnostic report generation on MIMIC-CXR with various metrics including BLEU-1/-2/-3/-4, ROUGE-L and METEOR.** The best-performing models are highlighted in red. The comparison methods include: METransformer (Wang et al., 2023b), DCL (Li et al., 2023b), R2GenGPT (Wang et al., 2023c), PromptMRG (Jin et al., 2024), BtspLLM (Liu et al., 2024a), CPO (Yang et al., 2025b) and MambaXray (Wang et al., 2025b).

| | Venue | BLEU-1 | BLEU-2 | BLEU-3 | BLEU-4 | ROUGE-L | METEOR |
|---|---|---|---|---|---|---|---|
| METransformer | CVPR'23 | 0.39 | 0.25 | 0.17 | 0.12 | 0.29 | 0.15 |
| DCL | CVPR'23 | - | - | - | 0.11 | 0.28 | 0.15 |
| R2GenGPT | MetaRad'23 | 0.41 | 0.26 | 0.17 | 0.13 | 0.29 | 0.17 |
| PromptMRG | AAAI'24 | 0.40 | - | - | 0.11 | 0.27 | 0.16 |
| BtspLLM | AAAI'24 | 0.40 | 0.26 | 0.18 | 0.13 | 0.29 | 0.18 |
| CPO | NeurIPS'25 | 0.43 | 0.29 | 0.19 | 0.16 | 0.42 | 0.29 |
| MambaXray | CVPR'25 | 0.42 | 0.27 | 0.18 | 0.13 | 0.29 | 0.17 |
| **Ours** | **This paper** | **0.56** | **0.37** | **0.27** | **0.19** | **0.30** | **0.21** |

*Table 2.* **Evaluation results of multiple source MLLM on classification of MS-CXR-T for comparison.** Top-1 accuracy is applied to evaluate the performance of different methods. The best-performing models are highlighted in red, with the second-best in blue. The comparison MLLMs includes: Claude Sonnet-4 (Anthropic, 2025), Gemini-2.5 (Comanici et al., 2025), GLM-4.5V (Team et al., 2025), GPT-5 (OpenAI, 2025), Qwen-VL-Max (Bai et al., 2025), Grok-4 (xAI, 2025) and Moonshot-v1 (AI, 2025).

| MLLMs | Con. | PE | Pna. | Pnx. | Ede. | Avg. |
|---|---|---|---|---|---|---|
| *Source MLLMs with Huge Parameters* | | | | | | |
| GPT-5 | 0.75 | 0.68 | 0.89 | 0.90 | 0.52 | 0.75 |
| Gemini-2.5 | 0.28 | 0.61 | 0.40 | 0.94 | 0.42 | 0.53 |
| Sonnet-4 | 0.89 | 0.69 | 0.48 | 0.15 | 0.15 | 0.47 |
| Qwen-VL-Max | 0.54 | 0.65 | 0.40 | 0.95 | 0.64 | 0.64 |
| Grok-4 | 0.43 | 0.41 | 0.36 | 0.97 | 0.61 | 0.56 |
| GLM-4.5V | 0.59 | 0.67 | 0.52 | 0.96 | 0.72 | 0.69 |
| Moonshot | 0.13 | 0.46 | 0.77 | 0.88 | 0.19 | 0.48 |
| *Target Model (7B MLLM)* | | | | | | |
| **Ours** | **0.84** | **0.67** | **0.78** | **0.96** | **0.65** | **0.78** |

shoulders of giants.

A closer examination of individual pathologies reveals the robustness of our approach against inter-model drift. While the target model does not strictly surpass the single best-performing specialist for every disease, it exhibits superior stability, consistently securing the second-best performance across nearly all categories. This stability is particularly critical in scenarios characterized by extreme inter-model divergence, such as Consolidation (Con.) and Edema (Ede.), where accuracy gaps among source models exceed 0.60. In these high-drift regimes, the target model acts as a robust stabilizer. By treating divergence as negative constraints, our framework avoids the catastrophic variance observed in individual sources, such as Moonshot's collapse to 0.13 on Con. or Sonnet-4's drop to 0.15 on Ede., thereby preventing

biased knowledge from infiltrating the reasoning process.

### 3.2. Harmonious Thinking Consistency

Beyond mere classification, we further substantiate the superiority of our framework in fostering consistent reasoning, which preserves beneficial CoT patterns across multiple MLLMs while effectively mitigating conceptual drift. To evaluate the target model's reasoning robustness and clinical narrative quality, we conduct diagnostic report generation tasks on the MIMIC-CXR dataset. As reported in Table 3, we employ a comprehensive suite of metrics: BLEU-n to quantify terminology precision and reasoning coherence, ROUGE-L to assess narrative structural completeness, and METEOR to capture synonym-aware semantic alignment.

The empirical results demonstrate that our framework consistently outperforms state-of-the-art methods across all dimensions. Notably, as shown in Table 3, our model achieves significant performance leaps, reaching 0.19 in BLEU-4 and 0.21 in METEOR. These gains specifically reflect a higher degree of reasoning consistency and lexical alignment precision, proving that our model does not merely mimic teacher outputs but internalizes a more accurate medical logic.

### 3.3. Generalized Reasoning Alignment

To further examine the cross-domain adaptability of our framework, we evaluate its generalized reasoning alignment through zero-shot multi-label classification across four rigorous benchmarks. As illustrated in Table 4, our model consistently outperforms the state-of-the-art competitive baselines, such as CARZero and CPO. Notably, we achieve superior AUC scores of 0.85 on Open-I and 0.83 on ChestXray14, underscoring the model's capacity to maintain precise conceptual grounding in unseen clinical scenarios.

Furthermore, we conduct a comparative analysis against contemporary reasoning alignment strategies on the MS-CXR-T dataset. The results in Table 5 demonstrate that our

*Table 4.* **Evaluation results of zero-shot diseases classification on Open-I (Demner-Fushman et al., 2012), ChestXray14 (Xray14) (Wang et al., 2017) , ChestXpert (Xpert) (Irvin et al., 2019) and ChestXDet10 (XDet10) (Liu et al., 2020).** AUC is applied to evaluate the performance of different methods. The best-performing models are highlighted in red. The comparison methods include: GLoRIA (Huang et al., 2021), MedCLIP (Wang et al., 2022), CheXzero (Tiu et al., 2022), BioViL (Bannur et al., 2023b), MedKLIP (Wu et al., 2023), KAD (Zhang et al., 2023), BiomedCLIP (Zhang et al., 2025b), CARZero (Lai et al., 2024) and CPO (Yang et al., 2025b).

| Method | Venue | Open-I | Xray14 | Xpert | XDet10 |
|---|---|---|---|---|---|
| GLoRIA | ICCV'21 | 0.59 | 0.61 | 0.75 | 0.65 |
| MedCLIP | EMNLP'22 | 0.55 | 0.56 | 0.74 | 0.57 |
| CheXzero | NBE'22 | 0.76 | 0.73 | 0.88 | 0.71 |
| BioViL | CVPR'23 | 0.70 | 0.73 | 0.79 | 0.71 |
| MedKLIP | ICCV'23 | 0.76 | 0.73 | 0.88 | 0.71 |
| KAD | NC'23 | 0.81 | 0.79 | 0.91 | 0.74 |
| BiomedCLIP | NEJM'24 | 0.58 | 0.64 | 0.68 | 0.63 |
| CARZero | CVPR'24 | 0.84 | 0.81 | 0.92 | 0.80 |
| CPO | NeurIPS'25 | 0.84 | 0.82 | 0.92 | 0.80 |
| **Ours** | **This paper** | **0.85** | **0.83** | **0.92** | **0.81** |

approach achieves an leading average score of 0.78, significantly surpassing methods like DistiLLM-2 and ABKD. This performance leap validates that our autonomous preference optimization does not merely mimic teacher behaviors but internalizes a more robust and transferable reasoning logic. By effectively filtering inconsistent signals, our framework ensures that the alignment remains resilient and generalized, even when transitioning from complex report generation to fine-grained disease classification.

*Table 5.* **Evaluation results with various reasoning alignment methods on multi-label chest diseases classification on MS-CXR-T.** The best-performing models are highlighted in red. The comparison methods include: $f$-Distill (Wen et al., 2023), GKD (Agarwal et al., 2024) MiniLLM (Gu et al., 2024), DistiLLM-2 (Ko et al., 2025) and ABKD (Wang et al., 2025a).

| | Venue | Con. | PE | Pna. | Pnx. | Ede. | Avg. |
|---|---|---|---|---|---|---|---|
| $f$-Distill | ACL'23 | 0.22 | 0.57 | 0.85 | 0.95 | 0.31 | 0.58 |
| GKD | ICLR'24 | 0.78 | 0.62 | 0.70 | 0.94 | 0.58 | 0.72 |
| MiniLLM | ICLR'24 | 0.77 | 0.56 | 0.78 | 0.94 | 0.61 | 0.73 |
| DistiLLM-2 | ICML'25 | 0.82 | 0.62 | 0.78 | 0.95 | 0.60 | 0.75 |
| ABKD | ICML'25 | 0.83 | 0.56 | 0.77 | 0.96 | 0.63 | 0.75 |
| **Ours** | | **0.84** | **0.67** | **0.78** | **0.96** | **0.65** | **0.78** |

### 3.4. Generalization on Multi-modal Applications

In addition, we have further expanded multi-modal applications, conducting experiments in the field of autonomous driving on BDD-X dataset, following the MCAM (Cheng

*Table 6.* Evaluation results of multi-modal applications in the field of autonomous driving on BDD-X dataset (Kim et al., 2018), following the MCAM (Cheng et al., 2025) and Dolphins (Ma et al., 2024). BLEU-4, CIDEr and METEOR are leveraged to evaluate the reasoning performance of driving actions. The comparison methods include: ADAPT (Jin et al., 2023), DriveGPT4 (Xu et al., 2024), RAG-Driver (Yuan et al., 2024a), MCAM (Cheng et al., 2025) and Dolphins (Ma et al., 2024).

| Methods | Venue | BLEU-4 | CIDEr | METEOR |
|---|---|---|---|---|
| ADAPT | ICRA'23 | 11.2 | 107.2 | 15.0 |
| DriveGPT4 | RAL'24 | 9.4 | 102.7 | 14.6 |
| RAG-Driver | RSS'24 | 11.1 | 109.1 | 14.8 |
| Dolphins | ECCV'24 | 15.0 | 134.2 | 17.3 |
| MCAM | ICCV'25 | 9.1 | 94.1 | 14.0 |
| **Ours** | **This Paper** | **16.5** | **140.2** | **18.2** |

et al., 2025) and Dolphins (Ma et al., 2024). As presented in Table 6, our method significantly outperforms recent models like MCAM (Cheng et al., 2025) and Dolphins (Ma et al., 2024). Notably, our reasoning CIDEr of 140.2 shows a substantial gain over MCAM of 94.1. It demonstrates that our strategy of treating divergent MLLMs' trajectories as constraints is highly effective in complex driving scenarios, demonstrating the generalization of our proposed method in multi-modal applications.

### 3.5. Ablation Studies

#### 3.5.1. ABLATION STUDIES ON MODULES

Moreover, we conduct ablation experiments on MIMIC-CXR to validate the feasibility and coordination of the multiple MLLMs (MT) and autonomous preference optimization (APO) under non-stationary environments, as presented in Table 7. Moreover, since APO inherently relies on concept alignment across multiple MLLMs, we do not conduct an ablation study of APO under the setting where MT is absent.

*Table 7.* **Ablation evaluation results on supervised pre-distillation (SPD), multiple teachers (MT) and autonomous preference (APO) under non-stationary distillation on MIMIC-CXR.** The ✓ denotes that the results are trained with the corresponding module. The results are based on the test split of the MS-CXR-T with the Top-1 accuracy.

| SPD | MT | APO | Con. | PE | Pna. | Pnx. | Ede. | Avg. |
|---|---|---|---|---|---|---|---|---|
| ✓ | - | - | 0.78 | 0.58 | 0.70 | 0.95 | 0.31 | 0.66 |
| ✓ | ✓ | - | 0.77 | 0.49 | 0.69 | 0.94 | 0.51 | 0.68 |
| ✓ | ✓ | ✓ | 0.84 | 0.67 | 0.78 | 0.96 | 0.65 | 0.78 |

The ablation on MT reveals only marginal overall gains, while performance on most diseases, including Con., PE,

Pna. and Pnx. deteriorates. This corroborates our observation that the unpredictable drift among source MLLMs severely disrupts the target's learning and degrades its effectiveness. Besides, compared with MT and SPD, APO delivers significant accuracy gains across all diseases by blocking the transmission of concept drift and enabling the target model to constructively learn all source MLLMs. Thus, the consistent, robust, and generalizable improvements confirm that the performance boost arises from APO itself rather than MT.

### 3.5.2. ABLATION STUDIES ON APO

*Table 8.* **Ablation evaluation results on direct baseline of optimization strategy on MIMIC-CXR**. The results are based on the test split of the MS-CXR-T with the Top-1 accuracy.

| Optimization Strategy | Con. | PE | Pna. | Pnx. | Ede. | Avg. |
|---|---|---|---|---|---|---|
| SFT (GT) | 0.69 | 0.63 | 0.69 | 0.75 | 0.73 | 0.70 |
| SFT (Best Teacher) | 0.64 | 0.64 | 0.55 | 0.86 | 0.70 | 0.68 |
| SFT (Consensus) | 0.71 | 0.65 | 0.59 | 0.89 | 0.71 | 0.71 |
| Single-negative DPO | 0.70 | 0.62 | 0.60 | 0.90 | 0.73 | 0.71 |
| Pairwise DPO | 0.74 | 0.62 | 0.62 | 0.91 | 0.76 | 0.73 |
| Ours | 0.84 | 0.67 | 0.78 | 0.96 | 0.65 | 0.78 |

Furthermore, we also conduct ablation experiments on more direct baselines compared with our APO, as presented in Table 8. It includes SFT (supervised fine-tuning) with ground truth diagnostic report, SFT with GPT-5, the best performance teacher as shown in Table 2, SFT with only consensus extracted by SPD, Single-negative DPO and Pairwise DPO.

First, the leap from Consensus-only SFT to APO proves that negative samples, as constraints, can better tighten the boundaries of the model. Besides, APO outperforms both the single-negative variant and pairwise DPO. It demonstrates that ours suppresses the complex drift inherent in non-stationary multi-MLLMs. Moreover, our model outperforms SFT on the best single MLLM (GPT-5). It is verified that learning the consensus of multiple models yields better results than imitating the strongest model.

### 3.5.3. ABLATION STUDIES ON VARIOUS TEACHERS

*Table 9.* **Ablation evaluation results on varying the number of source MLLMs**. The results are based on disease classification on MS-CXR-T, with average accuracy (Avg. Acc.), and on report generation on MIMIC-CXR, with BLEU-4 and METEOR.

| Numbers | Avg. Acc. | BLEU-4 | METEOR |
|---|---|---|---|
| 3 (GPT, Grok, Sonnet) | 0.53 | 0.11 | 0.15 |
| 5 (Add Qwen-VL, Gemini) | 0.78 | 0.17 | 0.19 |
| 7 | 0.84 | 0.19 | 0.21 |

We conducted additional ablation studies by varying the

number of source MLLMs ($N$) to investigate the impact of MLLM quantity on alignment performance, as exhibited in Table 9. The results demonstrate a significant performance leap as $N$ increases. This confirms that APO effectively leverages the complementary divergence among source models rather than just imitating the best teacher. By incorporating models with varying accuracy levels, each additional teacher provides unique drifting trajectories that serve as critical negative constraints. A larger allows APO to more precisely delineate the robust consensus manifold by suppressing a wider variety of individual hallucinations.

## 4. Conclusions and Limitations

In this paper, we introduce the Autonomous Preference Optimization (APO) for robust reasoning alignment in non-stationary environments. By formalizing inter-model drift as dynamic negative constraints, APO transforms alignment into a constraint satisfaction problem. Empirical results confirm that this paradigm effectively suppresses drift and synthesizes a robust consensus manifold from diverse sources, establishing a principled path for autonomous label-free model evolution.

We envision that this work will stimulate further progress in reasoning alignment for MLLMs, particularly in addressing domain-specific biases. Looking ahead, our future efforts will concentrate on enhancing the efficiency and reducing the computational cost in large-scale multimodal settings.

## Acknowledgment

The work was supported by the Australian Research Council (ARC) under Laureate project FL190100149.

## Impact Statement

This work adheres to the ICML Code of Ethics. In this study, no human subjects or animal experimentation was involved. All datasets used, including CXR-MAX, MIMIC-CXR, MS-CXR-T, Open-I, ChestXray14, ChestXpert and ChestXDet10, were sourced in compliance with relevant usage guidelines, ensuring no violation of privacy. We have taken care to avoid any biases or discriminatory outcomes in our research process. No personally identifiable information was used, and no experiments were conducted that could raise privacy or security concerns. We are committed to maintaining transparency and integrity throughout the research process.

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

# A. Related Works

## A.1. Concept Drift

Building on an extensive body of work, Lu et al. (Lu et al., 2019; 2020) provide a systematic survey that organizes concept drift mitigation into three dominant families: error rate–driven adaptation (Wang et al., 2024; Jiao et al., 2024), distribution-aware approaches (Yang et al., 2025a; Cerqueira et al., 2023; Yang et al., 2023), and multi-hypothesis frameworks (Yu et al., 2024; 2022; 2026; Shi et al., 2026). Our study is situated within the distribution-oriented stream, which is notable for coupling rigorous statistical tests with broad representational power, thereby enabling not only accurate detection of drift but also its nuanced characterization along temporal, spatial, and quantitative axes. By supporting fine-grained diagnostics such as the timing of drift onset, the attribution of drift to specific feature subspaces, and the assessment of its magnitude, distribution-based methods provide a principled foundation for adaptive systems that demand both interpretability and precise recalibration in the presence of evolving data.

Ongoing research on concept drift adaptation has produced a wide spectrum of refined techniques designed for increasingly complex learning environments. Among them, the Online Boosting Adaptive Learning (OBAL) framework (Yu et al., 2024) offers a two-stage pipeline for multistream classification, beginning with Adaptive Covariate Shift Adaptation (AdaCOSA) to capture evolving inter-stream correlations, and subsequently employing a Gaussian Mixture Model–driven weighting scheme to counter asynchronous distributional changes. In the multimodal landscape, CDMLLM (Yang et al., 2025a) and RCP ,(Yang et al., 2025c) highlights the susceptibility of vision–language models to drift-induced biases that arise during both pre-training and fine-tuning, and proposes a unified remedy that integrates T-distribution calibration for long-tailed scenarios with explicit out-of-distribution detection, thereby reinforcing alignment stability. Beyond single-stream settings, GDDM (Yu et al., 2023) contributes a distribution-free statistical mechanism for uncovering subtle group-level shifts in multi-stream data, relying on adaptive hypothesis testing to achieve robust detection. Anticipatory strategies have also been explored, most notably in DDG-DA (Li et al., 2022-06-28), which projects potential environmental evolution by coupling predictive factor analysis with synthetic data generation, creating a principled bridge between current observations and future distributional states. Complementing these supervised paradigms, STUDD (Cerqueira et al., 2023) introduces an unsupervised teacher–student discrepancy model that measures predictive consistency to flag drift without dependence on annotated labels, thereby reconciling sensitivity to distributional change with the practical limitations of real-world deployment.

## A.2. Reasoning Alignment for LLMs and MLLMs

Reasoning alignment has evolved into a central paradigm for synchronizing the capabilities of compact and specialized target models with advanced source systems. In the era of BERT, alignment primarily focused on representation compression (Sun et al., 2019; Jiao et al., 2020; Sanh et al., 2019). However, with the advent of Large Language Models (LLMs), the focus has shifted towards aligning reasoning processes, transferring not just probability distributions but also robustness, safety, and logical consistency. For instance, strategies like on-policy alignment (Agarwal et al., 2023; Yang et al., 2024b) enable models to refine their trajectories via self-correction, mitigating distribution mismatches during fine-tuning. Furthermore, alignment protocols have been integrated with instruction tuning (Wang et al., 2023a; Zhou et al., 2023) and preference optimization (Ouyang et al., 2022; Bai et al., 2022), establishing a foundation for enhancing LLM reasoning through structured feedback rather than mere imitation.

In multi-modal contexts, alignment becomes essential for bridging the semantic gap between visual perception and textual reasoning. Foundational works like CLIP (Radford et al., 2021) and FILIP (Yao et al., 2022) introduced contrastive alignment for multi-modal grounding. Building on this, generative frameworks such as BLIP-2 (Li et al., 2023a), LLaVA (Liu et al., 2023), and InstructBLIP (Dai et al., 2023) employ alignment strategies to synchronize visual encoders with large language decoders, ensuring that visual signals are correctly translated into coherent reasoning chains. Recent innovations, such as Align-KD (Feng et al., 2025a) and MoVE-KD (Cao et al., 2025b), further explore cross-modal alignment by distilling ensemble signals from diverse visual encoders, demonstrating the growing necessity for robust alignment mechanisms in complex MLLM architectures. This trend extends to high-stakes domains, where alignment ensures reliability in tasks like robotic surgery VQA (Chen et al., 2024) and medical diagnostics (Liu et al., 2025c; Yang et al., 2025e; Young & Xu, 2026).

A critical yet under-explored dimension is multi-source reasoning alignment, which aims to synthesize diverse capabilities from heterogeneous source models. Early explorations in computer vision (Son et al., 2021; Yuan et al., 2021) have inspired recent extensions to MLLMs. For example, Gu et al. (Gu et al., 2025) propose aligning with multiple MLLMs for out-of-context news detection, while continual learning frameworks (Chen et al., 2024) address alignment under streaming

data. However, recent benchmarks on alignment under distribution shift (Zhang et al., 2025a) reveal a significant challenge: source models in these settings often exhibit concept drift, providing biased or conflicting supervision. Most existing methods assume stationary source distributions, failing to address the dynamic inconsistencies inherent in multi-stream reasoning.

Beyond conventional alignment strategies, recent studies have explored trustworthy multi-view learning under distributional inconsistency and imperfect observations. Specifically, conformalized multi-view classification has been introduced to improve decision reliability through uncertainty calibration across heterogeneous views (Liu et al., 2024b). Subsequent work further investigated adaptive rejection mechanisms for enhancing the reliability of multi-view classification under ambiguous or low-confidence scenarios (Liu et al., 2025a), as well as robust testing-time adaptation strategies for trusted multi-view classification in real-world environments (Liu et al., 2025b).

Overall, the field is transitioning from static compression to dynamic reasoning alignment. Yet, current approaches largely overlook the non-stationary nature of drifting source models. These gaps motivate our proposed Autonomous Preference Optimization (APO), which explicitly reformulates multi-source alignment as a constraint satisfaction problem to resolve bias inheritance and effectively synthesize a robust consensus from drifting reasoning trajectories.

### A.3. Reinforced Fine-tuning in LLMs

The role of reinforcement learning (RL) in shaping post-training alignment of large language models (LLMs) has advanced significantly since OpenAI's pioneering work on Reinforcement Learning from Human Feedback (RLHF) (Christiano et al., 2017), which introduced a paradigm for aligning model behavior with human values (Ouyang et al., 2022). Initial implementations, such as OpenAI-o1 (Jaech et al., 2024), demonstrated the practical utility of preference-driven modeling, yet the reliance on large-scale human annotation quickly revealed severe limitations in cost and scalability. These constraints have spurred a transition toward automated reward construction using pre-trained systems, opening the door to a new generation of alignment methods. Bai et al.'s (Bai et al., 2022) constitutional framework, for example, relies on sparse natural language feedback as an indirect supervisory signal, while DeepSeek's research line illustrates a staged trajectory: beginning with a purely RL-based baseline (R0), and subsequently extending to the R1 system (Guo et al., 2025), which cycles between supervised fine-tuning and their GRPO optimization scheme (Shao et al., 2024). This cyclic design improved generalization capacity and marks a broader trend toward increasingly autonomous alignment pipelines that minimize human involvement while retaining robust performance.

Concurrently, alignment research has diversified through a range of novel paradigms that extend beyond the classical RLHF formulation. ReST (Gulcehre et al., 2023) advances iterative self-training by generating policy-driven samples and refining them via offline RL, while DPO (Rafailov et al., 2023) reconceptualizes the task as direct optimization of preferences through implicit reward modeling. Complementary efforts include Rejection Sampling Fine-Tuning (RSFT) (Yuan et al., 2024b), which augments supervised training with carefully filtered reasoning trajectories, and ReFT (Trung et al., 2024), which couples supervised fine-tuning initialization with PPO-based exploration to progressively expand reasoning capabilities. Extending these principles to multimodal contexts, Visual-RFT (Liu et al., 2025d) adapts GRPO-driven strategies for visual-language alignment under limited data regimes, whereas B-STaR (Zeng et al., 2025) introduces dynamic configuration mechanisms that balance exploration and exploitation for self-improving systems. Methodological innovation has also been paralleled by advances in evaluation: Qwen-Math-PRM (Zhang et al., 2025c) integrates Monte Carlo estimation with LLM-as-judge consensus, building a hierarchical framework that captures both stepwise reasoning fidelity and holistic solution quality. Along a similar line, ViLaM (Yang et al., 2026a; 2025d) leverages reinforcement learning for unsupervised visual grounding, tackling the challenges of open-world multimodal understanding.

## B. CXR-MAX Dataset

In this section, we showcase the samples utilized for training and validation in our study, generated by various MLLMs, with the image and ground truth of the radiology report. And the prompt we used to various MLLMs is:

*""This is a patient's chest DR image. The patient has been diagnosed with <diseases>. Can you find the basis for the diagnosis in the image?"*

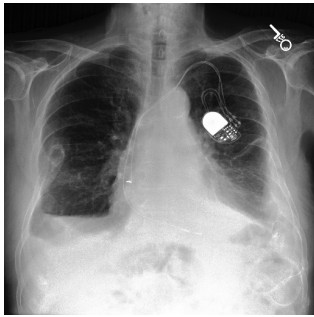

### Radiology Report

Findings: There is unchanged moderate left lower lobe atelectasis and bilateral pleural effusions. Linear atelectases in the left mid lung field are again seen, essentially unchanged. Lesion in the right seventh rib is unchanged. Pleurx catheter is seen, unchanged in position at the base of the left lung. There is no evidence of pneumothorax. Pacer is seen with leads appropriately placed within the right and left atria. Mediastinal silhouette is within normal limits with a calcified aorta.

### Claude Sonnet-4

Looking at this chest X-ray, I can identify several findings that support the diagnoses of atelectasis and pleural effusion:

**Evidence of Atelectasis:**

1. **Volume loss in the right hemithorax** - The right side appears smaller compared to the left

2. **Mediastinal shift** - The mediastinum (including the trachea and heart) appears shifted toward the right side, which is characteristic of right-sided volume loss

3. **Elevated right hemidiaphragm** - The right diaphragm appears higher than the left

4. **Increased opacity in the right lung** - Areas of collapsed lung tissue appear more dense/white

**Evidence of Pleural Effusion:**

1. **Blunting of the right costophrenic angle** - The normally sharp angle where the diaphragm meets the chest wall is obscured

2. **Increased density at the right lung base** - There appears to be fluid accumulation in the lower right pleural space

3. **Meniscus sign** - The fluid appears to have the characteristic curved upper border typical of pleural effusion

The combination of these findings - volume loss with mediastinal shift (atelectasis) and basal opacity with costophrenic angle blunting (pleural effusion) - provides clear radiographic evidence supporting both diagnoses. The presence of what appears to be a pacemaker or similar device in the right chest area is also noted.

### GPT-5-nano

Key findings that support atelectasis and pleural effusion:

• Left subclavian pacemaker/ICD in place (device visible in the left chest).

• Basilar opacities at the left lower lung suggesting subsegmental/plate-like atelectasis with some volume loss (diaphragm slightly elevated on the left).

• Blunting of the left costophrenic angle with posterior layering of fluid consistent with a small left pleural effusion (meniscus sign may be subtle on this view).

Right lung and mediastinal contours are otherwise relatively unremarkable.

In short: left basilar atelectasis with a small left pleural effusion. Correlate clinically; consider follow-up radiographs or ultrasound/CT if the clinical picture warrants.

---

GLM-4.5V

To determine the basis for diagnosing **Atelectasis** and **Pleural Effusion** in the chest X - ray, we analyze key imaging features:

**1. Pleural Effusion:**
Pleural effusion is identified by **blunting of the costophrenic angles** (the normally sharp, downward - pointing angles where the diaphragm meets the rib cage). In this image, the right costophrenic angle appears blurred and loses its typical sharp contour, which is a classic sign of fluid accumulation in the pleural space.

**2. Atelectasis:**
Atelectasis refers to partial lung collapse, which shows as **increased opacity (whiteness)** in the affected lung regions due to reduced air content. In this X - ray, there are areas of increased density in the lower lung fields (particularly on the right side), indicating alveolar collapse and loss of normal lung aeration.

In summary, the diagnostic clues are the **blunted right costophrenic angle (pleural effusion)** and **increased opacification in the lower lung zones (atelectasis)**.

The answer is The basis for the diagnosis includes blunted right costophrenic angle (indicating pleural effusion) and increased opacification in the lower lung zones (indicating atelectasis) .

---

Gemini-2.5-flash-lite

Based on the chest DR image, here are the findings that support the diagnoses of Atelectasis and Pleural Effusion:

**1. Pleural Effusion:**

- **Right Costophrenic Angle Blunting:** The right costophrenic angle is completely blunted and obscured by fluid.

- **Meniscus Sign:** There is a dense, homogeneous opacity at the right lung base with a characteristic curvilinear upper border (meniscus sign) that tracks upwards along the lateral chest wall. This indicates free fluid in the pleural space.

- **Obscured Diaphragm**: The right hemidiaphragm is completely obscured by the effusion.

This combination clearly indicates a significant right-sided pleural effusion.

**2. Atelectasis:**

- **Volume Loss in Right Hemithorax:** #blank

- **Mediastinal Shift:** The mediastinum, including the cardiac silhouette and potentially the trachea (though less clearly seen), is shifted towards the right (the affected side). This indicates a reduction in lung volume on the right.

- **Reduced Right Lung Volume:** The overall right lung field appears smaller and more collapsed compared to the left lung.

- **Increased Opacity/Consolidation:** The remaining lung parenchyma in the right hemithorax, particularly superior to the pleural effusion, appears denser and more opaque than normal lung, consistent with collapsed or consolidated lung tissue.

The atelectasis is likely, at least in part, compressive due to the large pleural effusion, but could also have an obstructive component.

In summary, the image demonstrates a large right-sided pleural effusion, causing blunting of the costophrenic angle and a meniscus sign, accompanied by signs of volume loss in the right hemithorax (mediastinal shift to the right) and increased density of the remaining lung, all consistent with a diagnosis of right-sided pleural effusion and atelectasis.

---

Moonshot-v1

As an AI, I'm not capable of providing medical diagnoses or interpreting medical images with the same level of expertise as a trained medical professional. However, I can provide general information on what to look for in a chest X-ray when atelectasis and pleural effusion are suspected.

Atelectasis is a condition where one or both lungs partially or completely collapse. On an X-ray, this might appear as an area of increased opacity (darker area) in the lung field, which could indicate that the lung tissue is not fully expanded.

Pleural effusion refers to the buildup of excess fluid in the pleural cavity, the space between the lungs and the chest wall. This can cause the lung to partially collapse and may appear on an X-ray as a layering density at the lung base or costophrenic angle, which could be seen as a meniscus sign (a curved line of increased density).

It's important to note that accurate diagnosis and interpretation of medical images should be done by a qualified radiologist or physician who can consider the full clinical context and other diagnostic information. If you have concerns about a medical condition, please consult a healthcare professional.

Qwen-VL-Max

The chest X-ray (DR image) shows several findings consistent with **atelectasis** and **pleural effusion**, particularly on the **left side**. Here's the basis for each diagnosis:

**1. Pleural Effusion (Left Side)**

**Key Findings:**

- **Blunting of the left costophrenic angle:** The sharp angle between the diaphragm and rib cage on the left is obscured or rounded, which is a classic sign of pleural fluid accumulation.

- **Homogeneous opacity in the lower left hemithorax:** There is increased density in the left lower lung field, especially near the base, which does not follow the bronchovascular markings. This suggests fluid rather than lung parenchymal disease.

- **Meniscus sign:** A concave upper border of the fluid collection (though less clearly visible here due to overlying structures), often seen in moderate pleural effusions.

- **Elevation of the left hemidiaphragm:** Fluid can push the diaphragm upward.

Note: The presence of a **cardiac pacemaker** on the left side may partially obscure the view, but the effusion is still evident.

**2. Atelectasis (Left Lower Lobe Likely)**

**Key Findings:**

- **Volume loss in the left lower lung:** The left hemithorax appears compressed, and there is a shift of mediastinal structures slightly toward the left (e.g., trachea and heart), indicating volume loss.

- **Loss of normal lung markings in the affected area:** The lung tissue appears denser and consolidated, with reduced air content.

- **Possible displacement of the minor fissure**: (if visible): In lobar atelectasis, the fissures may be displaced due to collapse.

The atelectasis likely results from **compression by the pleural effusion** —this is known as **compressive atelectasis**.

**Additional Notes:)**

- The **right lung** appears relatively clear, though some mild interstitial markings may be present, possibly due to chronic changes or early involvement.

- The **heart silhouette** appears enlarged, which could be due to the effusion pushing the heart laterally, or it may reflect underlying cardiomegaly.

- The **pacemaker device** is well-positioned, with leads extending into the right atrium and ventricle; no evidence of lead dislodgement.

**Summary:)**
☑ **Pleural effusion**: Left-sided, based on blunted costophrenic angle, homogeneous opacity, and diaphragmatic elevation.
☑ **Atelectasis**: Likely left lower lobe, secondary to compressive atelectasis from the adjacent pleural effusion.
These findings are typical of conditions such as pneumonia, heart failure, malignancy, or post-operative states, depending on clinical context.

**Recommendation**: Further evaluation with ultrasound or CT thorax may help confirm the nature and extent of the effusion and guide management (e.g., thoracentesis).

Grok-4

I can explain the basis for the diagnosis of atelectasis and pleural effusion based on an analysis of the chest X-ray (CXR) you provided. I'll keep this concise, objective, and focused on the visible findings in the image that support the diagnosis. Please note that I am not a substitute for the patient's actual treating physician, and CXR interpretation should always be correlated with the full clinical picture (e.g., symptoms, physical exam, labs, and possibly additional imaging like a CT scan or ultrasound for confirmation). If this is your CXR or that of someone you know, please discuss it directly with a doctor for personalized advice.

**Key Observations in the Image:**

This is a frontal (posteroanterior or PA) view of the chest. The lungs, heart, mediastinum (central chest structures), diaphragm, and bony thorax are visible. There's also an incidental implanted cardiac device (e.g., pacemaker or defibrillator) on the left side with leads extending into the heart—this is unrelated to the diagnosis but worth noting as it can sometimes complicate interpretation.

**1. Basis for Pleural Effusion:**

**Location**: Primarily on the right side (which appears on the left side of the image, as CXRs are conventionally viewed as if you're facing the patient).

**Key Findings**:

- Increased opacity (whitish haziness) at the right lung base, obscuring the normal sharp outline of the right hemidiaphragm (the dome-shaped muscle separating the chest from the abdomen).

- Blunting or filling-in of the right costophrenic angle (the sharp angle where the diaphragm meets the chest wall). Normally, this angle is very sharp and black (air-filled); here, it's filled with a denser, hazy material, which is classic for fluid accumulation.

- A subtle "meniscus sign" (curved upper border of the opacity) at the right base, suggesting free-flowing fluid layering dependently due to gravity in this upright view.

- The opacity is homogenous (evenly dense) and respects the lung boundaries without air bronchograms (air-filled airways standing out against the density), which fits with fluid in the pleural space rather than within the lung tissue itself.

**Why This Indicates Pleural Effusion**: Pleural effusion is an abnormal collection of fluid between the lung and chest wall. In an upright CXR, it typically settles at the base of the lung, causing these exact changes. The amount here appears moderate (not massive, as the mediastinum isn't shifted much), but it's enough to cause visible changes. Common causes include infection, heart failure, malignancy, or inflammation, but the CXR alone doesn't specify the cause.

**2. Basis for Atelectasis:**

**Location**: Also primarily involving the right lower lobe (base of the right lung).

**Key Findings**:

- Loss of volume and increased density in the right lower lung zone, making it appear smaller and whiter compared to the left lung (which is more uniformly black and air-filled).

- Crowding or absence of normal lung markings (vascular structures) in the affected area, with the remaining lung tissue appearing compressed.

- Subtle elevation or silhouetting of the right hemidiaphragm, suggesting collapse of the adjacent lung tissue.

- No obvious shift of the trachea or mediastinum (which would indicate more severe collapse), but the changes are localized to the base.

**Why This Indicates Atelectasis**: Atelectasis is partial or complete collapse of lung tissue, leading to reduced air content and increased density on CXR. It often occurs adjacent to a pleural effusion (the fluid compresses the lung), which is likely the case here—the effusion is probably contributing to compressive atelectasis. Other signs like air bronchograms are not prominent, suggesting it's not primarily consolidative (e.g., from pneumonia).

**3. Other Supporting Details:**

- **Bilateral Comparison**: The left lung appears relatively normal (darker and more aerated), which highlights the asymmetry and makes the right-sided abnormalities more evident.

- **No Contradictory Findings**: There are no signs of other issues (e.g., pneumothorax, large masses, or fractures) that might mimic these changes.

- **Technical Quality**: The image is well-exposed and positioned, reducing the chance of artifacts mimicking disease.

## C. Implementation Details

In this section, implementation details are provided.

In terms of the supervised fine-tuning progress, the hyperparameters are presented in Table 10. Qwen2.5-VL (7B) (Bai et al., 2025) is applied as our pre-trained model. During the SPD, we utilize the AdamW optimizer, which is configured

with a cosine annealing schedule as the learning policy. The initial learning rate is set to $1 \times 10^{-4}$, and the AdamW optimizer is employed with hyperparameters $\beta = (0.9, 0.98)$. Additionally, we set the weight decay to 0.05 and the dropout rate to 0.1. During the first 20 warm-up steps, the learning rate increases to $1 \times 10^{-4}$, and subsequently decays to $10^{-7}$. Unless otherwise specified, the supervised pre-distillation of our multi-modal large language model consists of 10,686 steps, executed on $2 \times 2$ NVIDIA A100 GPUs.

*Table 10.* The training hyperparameters of our MLLM.

| Supervised Pre-Distillation | |
|---|---|
| Training Steps | 10,686 |
| Warmup Steps | 20 |
| Warmup Ratio | 0.05 |
| Optimizer | AdamW |
| Learning Rate | 1e-4 |
| Learning Rate Decay | Cosine |
| Adam $\beta$ | (0.9, 0.98) |
| Weight Decay | 0.05 |
| Batch Size | 2 |

| Autonomous Preference Optimzation | |
|---|---|
| Training Steps | 12,132 |
| Warmup Steps | 0 |
| Optimizer | AdamW |
| Learning Rate | 2e-5 |
| Learning Rate Decay | Cosine |
| Adam $\beta$ | (0.9, 0.98) |
| Weight Decay | 0.05 |
| Batch Size | 2 |

While in the autonomous preference optimization (APO), the initial learning rate is reduced to $2 \times 10^{-5}$ without the warmup, with the batch size of 2. The visual encoder and text decoder are frozen out of the training. The reinforced custom-tuning consists of 12,132 steps, executed on $2 \times 2$ NVIDIA A100 GPUs. Other training parameters are the same as the fine-tuning.

