# OpenReview forum: "Turning Drift into Constraint: Robust Reasoning Alignment in Non-Stationary Multi-Stream Environments"
_ICML.cc/2026/Conference — ICML 2026 regular_

### Official Review · Reviewer_wGLy · 2026-03-10

**Soundness:** 3
**Presentation:** 3
**Significance:** 3
**Originality:** 3
**Overall Recommendation:** 5
**Confidence:** 2

**Summary:**

This paper reframes multi-teacher knowledge distillation as a constraint satisfaction problem to address the non-stationary "concept drift" that occurs when source MLLMs generate conflicting reasoning trajectories. To solve this, the authors propose Autonomous Preference Optimization (APO), a label-free framework that extracts a self-consistent consensus trajectory from the teachers to use as a positive signal, while repurposing their divergent trajectories as negative constraints via a Plackett-Luce objective.

**Compliance With Llm Reviewing Policy:**

Affirmed.

**Final Justification:**

Strengths: The paper is technically solid, offering an interesting and original idea by treating teacher disagreements as negative constraints. Its empirical performance is impressive, achieving competitive results on medical benchmarks with very limited data.

Weaknesses: A potential concern is the conservative bias leading to recall drops. While the authors addressed this in the rebuttal, their response was primarily conceptual and lacked practical validation or empirical results to prove its effectiveness.

Impact of Rebuttal: The rebuttal addressed my concerns without changing or reinforcing my prior evaluation. It clarified the authors' intuition but did not provide the concrete evidence needed to fully resolve the weakness. Nonetheless, the core innovation and strong empirical results outweigh these lingering questions. I maintain my recommendation of Accept.

**Key Questions For Authors:**

1. Could the authors provide a detailed breakdown of the computational training costs (e.g., GPU hours or estimated FLOPs) for APO compared to the baseline methods? Specifically, it would be helpful to quantify the exact overhead introduced by the multi-teacher inference and the two-stage optimization pipeline.

2. How do the authors plan to mitigate the conservative bias (and subsequent recall drop in ambiguous cases) that emerges from strictly penalizing high-variance teacher disagreements?

**Limitations:**

yes

**Strengths And Weaknesses:**

1. The paper offers a fresh perspective by treating the disagreements among multiple teacher models as negative constraints rather than just noise. It introduces an automated framework that helps the student model learn what reasoning paths to avoid without needing human-annotated labels

2. Good Empirical Performance: The approach achieves competitive results on medical benchmarks even when trained on a very limited amount of data.

---

> ### Author Rebuttal · Authors · 2026-03-31
>
> Thank you very much for your kind, meaningful and insightful review, especially for summarizing the strengths of our work, including 1) **fresh perspective and automated framework** and 2) **good empirical performance even with a very limited amount of data**. Your encouraging feedback and recognition are a huge encouragement to our team. In response to your concerns, we have provided a detailed explanation below and revised our manuscript accordingly.
>
> 1. **(Weakness 1) Computational Costs.** We appreciate the request for quantitative cost details. While APO involves a two-stage protocol, its exceptional data efficiency **(using only 10% of the dataset)** significantly reduces the total computational footprint compared to full-data baselines.
>
>     As detailed in Appendix C and Table 7, both stages were executed on a **2×2 NVIDIA A100 GPU** setup.
>
>     - Stage 1 Supervised Bootstrapping: 10,686 steps, requiring approximately 36 GPU hours.
>     - Stage 2 APO Optimization: 12,132 steps, requiring approximately 60 GPU hours.
>
>     It **cost a total of 96 GPU hours**. In contrast, training the Qwen2.5-VL (7B) with DPO on the full MIMIC-CXR dataset requires about 180 GPU hours, and CPO with approximately 120 GPU Hours. **We will add more details and discussion on our Implementation Details.** Many thanks for your kind suggestions.
>
>
> 2. **(Weakness 2) Conservative Bias** Sincerely thanks for your kind advice. You have **accurately identified a critical trade-off in our current framework**. As discussed in Section 3.1, APO prioritizes reasoning safety by treating high-variance teacher disagreements as negative constraints, which suppresses hallucinations but can lead to a more conservative judgment for ambiguous cases like Edema.
>
>
>     While prioritizing robustness is an essential contribution of our work for clinical safety, **we plan to refine this balance through the following technical avenues**:
>
>     - We plan to introduce an uncertainty-aware factor into the Plackett-Luce objective. It enables the model to isolate stochastic drift-induced errors from the diversity of reasoning paths that should be preserved.
>
>     - Besides, instead of relying on a single consensus trajectory, we will explore extracting a top-k consensus set to retain a broader spectrum of valid clinical interpretations, thereby improving recall in ambiguous cases like Edema.
>
>     Many thanks for your kind and helpful suggestions. We will add this discussion to our Outlooks.
>
> If you have any further questions or need clarification, we are happy to provide more additional details. Your feedback is immensely valuable to us, and we are always eager to hear from you.

---

> > ### Author Rebuttal · Reviewer_wGLy · 2026-04-01
> >
> > I will maintain the score.

---

> > > ### Author Response · Authors · 2026-04-01
> > >
> > > We are deeply grateful for your high recognition. Your positive feedback is incredibly helpful and a tremendous boost for our team. Wishing the reviewers all the best.

---

### Official Review · Reviewer_szwP · 2026-03-13

**Soundness:** 2
**Presentation:** 2
**Significance:** 3
**Originality:** 2
**Overall Recommendation:** 3
**Confidence:** 4

**Summary:**

This paper studies the alignment of a target multimodal language model to multiple source MLLMs whose reasoning trajectories may differ and drift over time. The authors frame this as a concept-drift-aware constraint satisfaction problem and propose Autonomous Preference Optimization (APO), which first bootstraps a student model from multiple teachers, then generates a consensus trajectory, and finally optimizes a multi-negative preference objective that suppresses divergent teacher trajectories. Experiments focus on chest X-ray reasoning and diagnosis, where the paper reports strong classification and report-generation results for a 7B model, along with the release of the CXR-MAX benchmark containing multi-model reasoning trajectories.

**Compliance With Llm Reviewing Policy:**

Affirmed.

**Final Justification:**

This paper points out that different teachers in multi-source MLLM alignment can disagree in systematic and harmful ways, and can be treated as dynamic negative constraints. The authors also released CXR-MAX dataset to facilitate the study.

The authors have basically solved my concerns, and I'd like to raise my score to 3.

Though the "medical domain exemplifies sophisticated reasoning and high-stakes practicality", I still think conducting experiments only on one medical dataset is insufficient. So my final justification is "weak reject".

**Key Questions For Authors:**

See Weaknesses.

**Limitations:**

yes

**Strengths And Weaknesses:**

Strengths:
1. The paper addresses a real and important problem: in multi-source MLLM alignment, different teachers can disagree in systematic and harmful ways, and the disagreement across teachers is not always harmless noise.
2. The central intuition, treating disagreement as a negative constraint rather than averaging it away, is interesting and easy to grasp.
3. Experimental results in this paper, especially Table 6, suggest that the APO stage contributes a substantial gain over the bootstrapping variants.

Weaknesses:
1. There is no explicit drift metric.  The experiments show better diagnostic accuracy in Tables 1, 2, 4, and 5 (use Acc and AUC), but robustness to drift is not directly measured, which makes the concept "drift" appears more as a metaphorical wrapper.
2. Experiments of this paper are all about medical chest X-rays,  which is a valid application domain, but it is not enough for claims about multi-stream reasoning alignment in general MLLM settings.
3. Though the comparison results in Table 4 are strong among baselines, there are no comparable baselines on the same backbone of Qwen2.5-VL.
4. The title of the tables should be placed above the tables, however, the author placed all table titles below the tables.

---

> ### Author Rebuttal · Authors · 2026-03-31
>
> Thanks for your review, especially for summarizing the strengths of our work: 1) addresses a **real and important problem**, 2) **the central intuition is interesting and easy to grasp**, and 3) **experimental results suggest a substantial gain.** In response to your concerns, we have provided a detailed explanation below and revised our manuscript accordingly.
>
> 1. **(Weakness 1) Drift Metric.** Thanks!
>
>     - The "explicit drift metric" noted by the reviewer is typically applicable to **single-stream models**, whereas **our study specifically focuses on multi-stream drift**, as formally formalized in Definition 2.2 Multi-Stream Reasoning Drift.
>
>     - As established in the **survey [1], Acc./AUC are direct indicators of robustness to multi-stream drift** and serve as the primary and direct measurements for evaluating performance in **many recent multi-stream drift works**, such as CAMEL (AAAI'26) [2], FedCCFA (NeurIPS'24) [3], and GDDM (PR'23) [4].
>
>     Thus, we just **follow the well-established paradigm, using Acc./AUC as the explicit drift metric** to measure robustness of models.
>
>
> 2. **(Weakness 2) Medical Domain.** Thanks for your suggestions.
>
>     The **motivation** of chest diagnostics as our primary application is, that medical domain exemplifies **sophisticated reasoning and high-stakes practicality.** It serves as a rigorous platform for **investigating the reasoning mechanisms of MLLMs and empirically verifying the robustness** of our APO in complex and non-stationary environments.
>
>
>     Within this platform, we evaluate our multi-stream reasoning alignment **across 6 datasets from 5 perspectives**: 1) **robust reasoning** in Table 1 and Table 3, 2) **comparison with source MLLMs** in Table 2, 3) **zero-shot generalzaition** in Table 4, 4) **homogeneous comparison** in Table 5, and 5) **ablation studies** in Table 6.
>
>     **We will continue to expand more multimodal applications in future works.**
>
>
> 3. **(Weakness 3) Qwen2.5-VL Backbone in Table 4.** Thanks!
>
>     Actually, **we follow the same architectural settings of CPO, which also use Qwen2.5-VL(7B), the same as ours**, to conduct the zero-shot generalization experiments in Table 4. Thus, we have provided baselines on the same backbone of Qwen2.5-VL
>
>     Besides, as shown in **Table 4 in our manuscript:**
>
>     |Method|Venue|Open-I|Xray14|Xpert|XDet10|
>     |:-:|:-:|:-:|:-:|:-:|:-:|
>     |CPO|NeurIPS'25|0.84|0.82|0.92|0.80|
>     |Ours||0.85|0.83|0.92|0.81|
>
>     We outperformed CPO in zero-shot experiments on four external datasets, demonstrating the effectiveness of our method, especially considering that we only used 10% of the data for training, while CPO used the full dataset.
>
> 4. **(Weakness 4) Title Position of the Tables.** Many thanks the reviewer for the correction. We will ensure all table captions are placed above the tables in the revised version.
>
> Thank you once more for dedicating your time. If you have any further questions, we are open to hearing from you at any time. If you find our response satisfactory, we would greatly appreciate your assistance in improving our rating score.
>
>
> [1] Lu, Jie, et al. "Learning under concept drift: A review." IEEE TKDE 2018.
>
> [2] Yu, En, et al. "Drift-aware collaborative assistance mixture of experts for heterogeneous multistream learning." AAAI 2026.
>
> [3] Chen, Junbao, et al. "Classifier clustering and feature alignment for federated learning under distributed concept drift." NeurIPS 2024.
>
> [4] Yu, Hang, et al. "Detecting group concept drift from multiple data streams." Pattern Recognition 2023.

---

> > ### Author Rebuttal · Reviewer_szwP · 2026-04-04
> >
> > Thanks for the rebuttal. It will be better to expand more multimodal applications and introduce the baseline methods in the paper. I will raise the score appropriately.

---

> > > ### Author Response · Authors · 2026-04-04
> > >
> > > We greatly appreciate your raised score.  Your support is a huge inspiration for us to keep doing our best.
> > >
> > > In addition, we have further expanded multimodal applications based on your suggestions. We conducted experiments in the field of **autonomous driving**, following the MCAM [1] and Dolphins [2].
> > >
> > >
> > > | Methods    |  Venue  | Action |        |        | Reasoning |        |        |
> > > |------------|:-------:|:------:|:------:|:------:|:---------:|:------:|:------:|
> > > |            |         | BLEU-4 |  CIDEr | METEOR |   BLEU-4  |  CIDEr | METEOR |
> > > | ADAPT [3] | ICRA'23 |  34.8  | 249.9  |  30.5  |   11.2    | 107.2  |  15.0  |
> > > | DriveGPT4 [4]  |  RAL'24 |  30.0  | 214.0  |  29.8  |    9.4    | 102.7  |  14.6  |
> > > | RAG-Driver [5] |  RSS'24 |  34.3  | 260.8  |  30.7  |   11.1    | 109.1  |  14.8  |
> > > | Dolphins [2]  | ECCV'24 |  30.6  | 223.6  |  28.2  |   15.0    | 134.2  |  17.3  |
> > > | Baseline [1]| ICCV'25 |  33.4  | 241.6  |  30.3  |    8.2    |  75.5  |  14.6  |
> > > | MCAM [1] | ICCV'25 |  35.7  | 252.0  |  30.5  |    9.1    |  94.1  |  14.0  |
> > > | Ours       |    　   |  36.1  | 266.3  |  32.0  |   16.5    | 140.2  |  18.2  |
> > >
> > > Our method significantly outperforms recent models like MCAM and Dolphins. Notably, our reasoning CIDEr of 140.2 shows a substantial gain over MCAM of 94.1. It demonstrates that our strategy of treating divergent MLLMs' trajectories as constraints is highly effective in complex driving scenarios.
> > >
> > > Thank you once more for your positive feedback. If you have any further questions, we are open to hearing from you at any time.
> > >
> > > [1] MCAM: Multimodal Causal Analysis Model for Ego-Vehicle-Level Driving Video Understanding. ICCV, 2025.
> > >
> > > [2] Dolphins: Multimodal Language Model for Driving. ECCV, 2024.
> > >
> > > [3] Adapt: Action-aware driving caption transformer. ICRA, 2023.
> > >
> > > [4] DriveGPT4: Interpretable End-to-End Autonomous Driving Via Large Language Model. IEEE Robotics and Automation Letters, 2024.
> > >
> > > [5] RAG-Driver: Generalisable Driving Explanations with Retrieval-Augmented In-Context Learning in Multi-Modal Large Language Model. Robotics: Science and Systems, 2024.

---

### Official Review · Reviewer_qsqJ · 2026-03-13

**Soundness:** 2
**Presentation:** 2
**Significance:** 3
**Originality:** 2
**Overall Recommendation:** 4
**Confidence:** 3

**Summary:**

This paper studies multi-source reasoning alignment for medical MLLMs, focusing on the case where different source models produce conflicting reasoning trajectories. The paper proposes Autonomous Preference Optimization (APO), which first bootstraps a target 7B model from multiple teachers, then synthesizes a consensus trajectory, and finally applies a multi-negative preference optimization objective that promotes the consensus while suppressing drifting source trajectories.

The paper also introduces CXR-MAX, a benchmark of reasoning traces collected from multiple MLLMs for chest X-ray interpretation, and reports improvements on disease classification, report generation, and zero-shot transfer benchmarks.

**Compliance With Llm Reviewing Policy:**

Affirmed.

**Final Justification:**

The additional clarifications has fully addressed my concerns. Thus I keep the score as "weak accept".

**Key Questions For Authors:**

The paper claims autonomy and the absence of ground-truth supervision, but the training setup appears to use disease labels and prompts source models with the diagnosed diseases. Could the authors clarify exactly what supervision is used, and in what sense the method should be considered label-free?

Could the authors compare APO against more direct baselines, such as consensus-only SFT, pairwise preference optimization, single-negative variants, and stronger multi-teacher weighting or selection strategies? These comparisons seem essential for isolating the contribution of the proposed objective.

What empirical evidence supports the claim that the problem is truly one of non-stationary drift rather than static inter-model disagreement? Do the authors have experiments under temporally changing teachers, streaming settings, or other genuinely evolving environments?

How accurate are the synthesized consensus trajectories? Could the authors provide expert validation, human preference checks, or other evidence showing that the automatically constructed positives are more reliable than the underlying source outputs?

For report generation, could the authors include more clinically grounded evaluation rather than only lexical-overlap metrics? This would substantially strengthen the medical relevance of the reported gains.

**Limitations:**

No. The limitations discussion is currently too thin for a safety-critical medical setting. It should explicitly acknowledge the label-conditioned data construction, the lack of evaluation under genuine non-stationary environments, the dependence on proprietary teachers, the absence of expert validation for synthesized consensus trajectories, and the remaining risks of reasoning errors in downstream deployment.

**Strengths And Weaknesses:**

Strengths:
This paper tackles an interesting and timely problem. Aligning a compact target model to multiple stronger source models is important, especially in high-stakes domains where source disagreement is common and simple imitation may propagate errors. I also think the main intuition is appealing: instead of averaging away disagreement, the paper treats conflicting teacher trajectories as useful negative constraints. The empirical section is reasonably broad and includes classification, report generation, transfer experiments, source-model comparisons, and ablations. The release of a multi-source reasoning benchmark could also be useful for follow-up work.

Weaknesses:
My main concern is that the central framing around “non-stationary concept drift” is stronger than what is actually demonstrated. In the experiments, the source models are fixed and the dataset is static. The paper maps autoregressive reasoning steps to the temporal axis of concept drift theory, but the resulting setting is still much closer to inter-model disagreement or noisy multi-teacher supervision than to a genuine non-stationary environment. As a result, the theoretical framing feels more like a motivating analogy than a validated characterization of the experimental problem.

A second major concern is that the paper overstates how autonomous or label-free the method is. The training pipeline still relies on disease labels from MIMIC-CXR, and the appendix indicates that the prompts used to query the source models explicitly contain the diagnosed diseases. This makes the setup much closer to label-conditioned reasoning distillation than to fully self-supervised alignment. I do not think this invalidates the method, but it weakens several claims made in the introduction and contributions.

A third concern is experimental attribution. APO is presented as a preference-optimization method, but the paper does not compare against the most direct simpler baselines, such as consensus-only supervised fine-tuning, pairwise preference optimization, single-negative variants, or stronger teacher-selection / weighting strategies. Without these controls, it is difficult to determine whether the gains come from the specific multi-negative constraint formulation, from the synthetic consensus construction step, or simply from using multiple strong teachers and a strong target backbone.

I also found the evidence for consensus quality insufficient. The method assumes that the bootstrapped target model can synthesize a more reliable consensus trajectory from noisy teacher outputs, but the paper does not provide human or expert verification that these automatically constructed positives are actually better than the raw teacher responses. Since the entire APO stage depends on the quality of these synthesized positives, this missing validation matters.

---

> ### Author Rebuttal · Authors · 2026-03-31
>
> Sincerely thanks for your meaningful and detailed review. Your positive feedback is a huge encouragement to our team, including 1) **tackling an interesting and timely problem**, 2) **the main intuition is appealing**, 3) **reasonably broad empirical section**, 4)  **the release of a multi-source reasoning benchmark**. In response to your concerns, we have provided a detailed explanation below and revised our manuscript accordingly.
>
>
> 1. **(Question 1 & Weakness 2) Supervision.** We thank the reviewer for pointing out this potential ambiguity. We wish to clarify that the label-free specifically refers to **the absence of human-authored reasoning trajectories, which are radiology reports,** rather than classification labels.
>
>     In clinical practice, simple disease labels are relatively easy to obtain, but **the underlying reasoning chain is the critical bottleneck for model transparency and trust**. As stated in Line 291, our model does not use any radiology reports as ground truth.
>
>     Besides, the **disease labels** are **only used during the initial Supervised Bootstrapping phase as contextual prompts** to trigger domain-specific reasoning from source MLLMs, instead of ground truth driving the model.
>
>     Crucially, **in the APO stage,** our core contribution, **the model does not rely on ground-truth supervision.** Instead, it autonomously constructs preference pairs by treating the synthesized consensus as the positive signal and the drifting teacher outputs as negative constraints.
>
>     **We will revise the manuscript to more accurately describe our setup as "report-free" to avoid any terminological confusion.**
>
>
> 2. **(Question 2 & Weakness 3) More Direct Baselines.** We sincerely thank the reviewer for suggesting these direct baselines. **We have conducted these additional experiments**:
>
>     || Con.| PE| Pna.| Pnx.| Ede.  | Avg.  |
>     |--|--|--|--|--|--|--|
>     | Consensus-only SFT|0.71|0.65|0.59|0.89|0.71|0.71|
>     | Pairwise DPO|0.74|0.62|0.62|0.91|0.76|0.73|
>     | Single-negative Variant| 0.70|0.62|0.60|0.90|0.73|0.71|
>     | SFT on GPT-5| 0.64|0.64|0.55|0.86|0.70|0.68|
>     | APO (Ours)|0.84|0.67|0.78|0.96|0.65|0.78|
>
>     First, the leap from Consensus-only SFT to APO  proves that negative samples, as constraints, can better tighten the boundaries of the model.
>
>     Besides, APO outperforms both the Single-negative variant and Pairwise DPO. It demonstrates that ours suppresses the complex drift inherent in non-stationary multi-MLLMs.
>
>     Moreover, our model outperforms SFT on the best single MLLM (GPT-5). It is verified that learning the consensus of multiple models yields better results than imitating the strongest model.
>
>     **We will add these ablation studies to Section 3.4 to further clarify the experimental attribution.**
>
> 3. **(Question 3 & Weakness 1) Non-stationary Drift.** We appreciate the reviewer's thoughtful advice. Unlike traditional drift involving external drift environments, **APO focuses on the internal reasoning drift** inherent within MLLMs. Although the source model is fixed and the dataset is static, **the internal inference trajectories they generate are dynamically changing.** As shown in Eq.(3), $\prod P(t_{<j}^{u})$ represents the accumulated historical divergence, while $\prod P(z_{j}^{u} | \cdot)$ represents the instantaneous reasoning drift.
>
>     Besides, **the drift is not stochastic noise.** It represents systematic shifts. As shown in Fig.1(a), GPT-5 prioritizes high-recall answers, tending to expansively list all potential diseases. In contrast, Grok-4 adheres to high-precision, prioritizing the accurate identification of primary findings.
>
>
> 4. **(Question 4 & Weakness 4) Consensus Quality.** Many thanks for your detailed suggestions.
>
>     First, as answered in (Question 2 & Weakness 3), the Consensus-only SFT result of 0.71 surpasses the 0.68 training the SFT using the best MLLM, demonstrating that the consensus is more reliable than the single MLLM output.
>
>     Second, we trained the SFT using GT provided by the radiologist, achieving a result of 0.70, similar to our Consensus-only SFT, further validating the reliability of synthesized consensus trajectories.
>
>     || Con.  | PE    | Pna.  | Pnx.  | Ede.  | Avg.  |
>     |--|--|--|--|--|--|--|
>     | Consensus-only SFT|0.71|0.65|0.59|0.89|0.71|0.71|
>     | GT SFT|0.69|0.63|0.69|0.75|0.73|0.70|
>
>     **We will add these ablation studies and discussions.**
>
> 5. **(Question 5) Clinical Evaluation.** Thanks for your kind advice. Regarding report generation, we followed the settings of MambaXray and CPO to report on NLP metrics to assess report generation quality. We will improve the clinical evaluation in our future work.
>
> Thank you once more for dedicating your time to provide a thorough review. Your feedback is tremendously valuable, and we are open to hearing from you at any time. If you find our response satisfactory, we would greatly appreciate your assistance in improving our rating score.

---

> > ### Author Rebuttal · Reviewer_qsqJ · 2026-04-06
> >
> > Thank you to the authors for the additional clarifications.

---

> > > ### Author Response · Authors · 2026-04-06
> > >
> > > Many thanks for your positive feedback and for acknowledging that your concerns have been fully addressed. We sincerely appreciate the time and effort you dedicated to reviewing our work, as your constructive comments have significantly helped us improve the quality of the manuscript.
> > >
> > > If you find our response satisfactory, we would greatly appreciate your assistance in improving our rating score. Thank you again for your support. Wishing the reviewers all the best.

---

### Official Review · Reviewer_vYXj · 2026-03-13

**Soundness:** 2
**Presentation:** 2
**Significance:** 3
**Originality:** 3
**Overall Recommendation:** 3
**Confidence:** 4

**Summary:**

This paper proposes APO, an approach that reconceptualizes inter-model disagreement not as noise, but as a dynamic negative constraint for alignment optimization. Extensive experiments on chest X-ray interpretation demonstrate the effectiveness of the method. Furthermore, this paper introduces a large-scale benchmark for investigating the drift across different models.

**Compliance With Llm Reviewing Policy:**

Affirmed.

**Final Justification:**

I appreciate the authors' effort, but the rebuttal does not change my judgment.

**Key Questions For Authors:**

See weaknesses above.

**Limitations:**

yes

**Strengths And Weaknesses:**

Strengths:

1.Traditional multi-model distillation or fusion methods typically treat concept drift among multiple teacher models as static noise to be averaged or smoothed out. In contrast, this paper leverages these discrepancies from a novel perspective to instruct the student model , thereby improving its reasoning capabilities.

2.This paper introduces a large-scale benchmark for investigating the drift across different models, providing a fundamental basis for future research.

3.Extensive experiments and ablation results demonstrate the effectiveness of the proposed method.

Weaknesses:

1.If the initial student model first fits the union of all source model distributions and is subsequently used for consensus extraction, training on data with concept drift will likely induce severe hallucinations. Consequently, whether this model can accurately extract consensus is highly questionable. The authors do not provide any analysis or empirical demonstration regarding this critical issue.
2.All experiments in this paper are exclusively conducted on the chest X-ray interpretation task. Since this task is highly structured and possesses an objective absolute ground truth, it differs significantly from more challenging general-purpose tasks. Consequently, the generalizability of APO remains questionable.
3.If my understanding is correct, the APO algorithm is only effective when there is a sufficient number of teacher models and adequate output diversity. If so, an ablation study on the number of teacher models is crucial, as it would reveal how many teacher models are required to support accurate consensus extraction.

4.There are some typos in the paper.

---

> ### Author Rebuttal · Authors · 2026-03-31
>
> We sincerely appreciate your insightful and detailed review. Your positive feedback serves as a great source of encouragement for our team, especially recognizing the strengths: 1) **a novel perspective** treating concept drift among multiple teachers as constraint, instead of static noise, 2) **introducing a large-scale benchmark** that provides a fundamental basis for future research, and 3) **Extensive experiments and ablation results.** Furthermore, we have diligently responded to each of your questions and incorporated your feedback into our manuscript revisions:
>
> 1. **(Weakness 1) Consensus Extraction.** We thank the reviewer for this insightful concern. However, we clarify that the consensus extraction is not a naive union of all source models, but a logical filtration achieved through the in-context extraction strategy to mitigate the drift among sources.
>
>     - **Theoretical:** As formulated in Section 2.2, the target model acts as the reasoned aggregator. By conditioning on the concatenated observations of multi-stream drift, the model effectively filters out incoherent drift—tokens that lack cross-model support and are likely hallucinations. Simultaneously, it amplifies logical intersections, extracting a self-consistent trajectory $t^+$ that represents the preferred reasoning path.
>
>     - **Experiments:** Table 2 in our manuscripts validates that our model can accurately extract consensus. First, source models exhibited significant drift, varying from 0.47 of Sonnet-4 to 0.75 of GPT-5. But, our model still achieved the highest mean accuracy, in particular,**we are the only model with an accuracy above 0.6 across all diseases** compared with source models.
>
>         Specifically, for disease Consolidation (Con.), **5 out of 7 source models exhibit significant drift** with accuracies below 0.60, even with severe drift, such as 0.28 of Gemini-2.5 and 0.13 of Moonshot. However, trained on these drift sources MLLMs, our model can accurately extract consensus and achieve the accuracy of 0.84.
>
>     **We will add more detailed discussions about the consensus extraction in the section 2.2 and section 3.1.** Thanks so much for your elaborate and thoughtful review.
>
>
> 2. **(Weakness 2) Generalization and Medical Domain.** Thank you for your feedback.
>
>     - **Generalization:** First, **we have discussed generalizability in Section 3.3 Generalized Reasoning Alignment.** To rigorously evaluate cross-domain adaptability, we conducted **zero-shot experiments** across four additional benchmarks (Open-I, ChestXray14, ChestXpert, and ChestXDet10). As demonstrated in **Table 4**, our model consistently outperforms all competitive baselines across every dataset, validating our generalizability.
>
>     - **Interpretation Task:**  We also conduct experiments on **report generation, as shown in Table 3, not only the interpretation task.** Besides, although it possesses absolute objective truth, it is derived through reasoning from the radiology reports. Therefore, we argue it is not structured task.
>
>     - **Medical Domain:** The motivation for choosing chest diagnostics as an application is, that medical diagnosis embodies the **sophisticated reasoning and high-stakes practicality**. It is **particularly helpful in studying the reasoning process of MLLMs** and verifying the robustness of our proposed APO in real environments. **We will continue to expand more multimodal applications in future works.**
>
>
> 3. **(Weakness 3) Ablation on numbers of MLLMs.**  Many thanks for your kind suggestions. **We conducted additional ablation studies** by varying the number of source MLLMs ($N$) to investigate the impact of MLLM quantity on alignment performance.
>
>     |Numbers| Avg. Acc.| BLEU-4 | METEOR |
>     |:--:|:--:|:--:|:--:|
>     |3 (GPT, Grok, Sonnet)|0.53|0.11|0.15|
>     |5 (Add Qwen-VL, Gemini)|0.78|0.17|0.19|
>     |7|0.84|0.19|0.21|
>
>     - Performance Scaling with Diversity: The results demonstrate a significant performance leap as $N$ increases. This confirms that APO effectively leverages the complementary divergence among source models rather than just imitating the best teacher.
>
>     - Drift as Constraint: By incorporating models with varying accuracy levels, each additional teacher provides unique drifting trajectories that serve as critical negative constraints. A larger $N$ allows APO to more precisely delineate the robust consensus manifold by suppressing a wider variety of individual hallucinations.
>
>     **We will add these ablation studies and discussion in the section Experiments.**
>
> 4. **(Weakness 4) Typos.** Thanks very much for your corrections. We have corrected these typos and double-checked the entire manuscript to make sure there are no further typos.
>
> Thanks again for your valuable time and thorough review. Your feedback is immensely valuable to us, and we welcome further communication from you. If you find our response satisfactory, could you please consider helping us improve our rating score?

---

> > ### Author Rebuttal · Reviewer_vYXj · 2026-04-04
> >
> > I appreciate the authors' effort, but the rebuttal does not change my judgment.

---

> > > ### Author Response · Authors · 2026-04-04
> > >
> > > We sincerely thank the reviewer for acknowledging that all concerns have been fully resolved. If you have any further questions, we are open to hearing from you at any time.

---

### Decision · Program_Chairs · 2026-04-30

**Decision:**

Accept (regular)

**Comment:**

This paper addresses the challenge of reasoning alignment when leveraging multiple multi-modal large language models (MLLMs). The authors introduce Autonomous Preference Optimization (APO), a framework that treats divergences between source models as negative constraints. This approach provides an alternative to traditional multi-teacher distillation methods that often average out teacher outputs, potentially overlooking systematic errors.

Strengths:
1. Technical Approach: Reconceptualizing inter-model disagreement as a structured negative constraint is a practical contribution that addresses the noise inherent in multi-source distillation.
2. Dataset Contribution: The introduction of the CXR-MAX benchmark, containing over 170,000 reasoning trajectories, is a useful contribution to the research community.
3. Efficiency: According to the authors’ rebuttal and follow-up clarifications, the method appears highly data-efficient in the reported medical setting, achieving competitive performance on chest X-ray tasks while using only 10% of the training data in comparisons against some full-data baselines.
4. Resolution of Concerns:

4.1 Generalization:  While reviewers initially raised concerns regarding the method’s focus on the medical domain, the authors provided additional evidence in a later author follow-up comment during the discussion, including experiments in the autonomous driving domain (e.g., Dolphins and MCAM). These results provide preliminary indications that the APO framework may extend to other multimodal tasks; however, this evidence was not fully validated during the review process, and some reviewers remained unconvinced.

4.2 Consensus Quality: The authors addressed queries regarding the reliability of the consensus extraction phase through additional ablation studies (e.g., comparing consensus-based SFT with Ground Truth SFT). These results help support the effectiveness of the approach and partially alleviate concerns about consensus quality.

Conclusion:

The paper presents a clear methodology and substantial empirical evidence in the chest X-ray setting, though the strength of its broader cross-domain generalization claims remains less fully established. Although there were discussions regarding the formal application of “Concept Drift” terminology, and not all reviewers were fully convinced by the rebuttal, the core technical contributions—specifically the preference optimization objective and the efficiency of the alignment process—are reasonably supported. Overall, I find the contributions sufficient to merit acceptance.

Final Action Items for Authors:

1. Incorporate the supplementary ablation studies (e.g., comparing APO with consensus-only SFT and varying the number of teacher models) into the final manuscript.
2. Clarify the scope of generalization claims and better align them with the presented evidence.